# DNA Damage Tolerance Mechanisms Revealed from the Analysis of *Immunoglobulin V* Gene Diversification in Avian DT40 Cells

**DOI:** 10.3390/genes9120614

**Published:** 2018-12-07

**Authors:** Takuya Abe, Dana Branzei, Kouji Hirota

**Affiliations:** 1Department of Chemistry, Graduate School of Science and Engineering, Tokyo Metropolitan University, Minamiosawa 1-1, Hachioji-shi, Tokyo 192-0397, Japan; 0330abe@tmu.ac.jp; 2IFOM, the FIRC Institute of Molecular Oncology, Via Adamello 16, 20139 Milan, Italy; dana.branzei@ifom.eu; 3Istituto di Genetica Molecolare, Consiglio Nazionale delle Ricerche (IGM-CNR), Via Abbiategrasso 207, 27100 Pavia, Italy

**Keywords:** homologous recombination, translesion DNA synthesis, replication, activation-induced deaminase, abasic site, DNA damage

## Abstract

DNA replication is an essential biochemical reaction in dividing cells that frequently stalls at damaged sites. Homologous/homeologous recombination (HR)-mediated template switch and translesion DNA synthesis (TLS)-mediated bypass processes release arrested DNA replication forks. These mechanisms are pivotal for replication fork maintenance and play critical roles in DNA damage tolerance (DDT) and gap-filling. The avian DT40 B lymphocyte cell line provides an opportunity to examine HR-mediated template switch and TLS triggered by abasic sites by sequencing the constitutively diversifying immunoglobulin light-chain variable gene (*IgV*). During *IgV* diversification, activation-induced deaminase (AID) converts dC to dU, which in turn is excised by uracil DNA glycosylase and yields abasic sites within a defined window of around 500 base pairs. These abasic sites can induce gene conversion with a set of homeologous upstream pseudogenes via the HR-mediated template switch, resulting in templated mutagenesis, or can be bypassed directly by TLS, resulting in non-templated somatic hypermutation at dC/dG base pairs. In this review, we discuss recent works unveiling *IgV* diversification mechanisms in avian DT40 cells, which shed light on DDT mode usage in vertebrate cells and tolerance of abasic sites.

## 1. Introduction

Cellular DNA is continuously damaged by chemical and physical agents from both endogenous metabolic processes and exogenous insults. Replicative DNA polymerases replicate genomic DNA with extraordinarily high accuracy, making only a single error per 10^6^ nucleotides synthesized in vivo [1]. Due to this enzymatic property, replicative polymerases cannot accommodate nucleotides at damaged templates and thus arrest replication [2]. Cells employ several mechanisms for releasing arrested replication forks depending on the type of DNA lesions or the cell cycle phase [3]. One of the most common and formidable lesions is the abasic site. Abasic sites are estimated to occur at a frequency of over 10,000 sites per cell per day in mammalian cells [4,5]. These lesions can be repaired via accurate base excision repair [6,7,8,9]. However, when the replication fork encounters unrepaired abasic sites, they are dealt with by DNA damage tolerance (DDT) mechanisms that allow completion of replication beyond the damaged template and prevent formation of deleterious double strand breaks (DSBs) [10,11]. In general, there are two modes of DDT conserved in eukaryotes [10]. One mode is mediated through homologous recombination (HR), which mediates continuous replication using a newly synthesized sister strand [10,12,13] (Figure 1A). A second mode is translesion DNA synthesis (TLS), which employs specialized DNA polymerases, including polymerase η and polymerase ζ, to permit continuous replication beyond the damaged template [14,15,16,17,18] (Figure 1B).

When the blocking lesions are abasic sites, DDT processes serve additional functions, such as generating immunoglobulin diversity in B cells [19,20]. Avian B cells diversify the *immunoglobulin variable* (*IgV*) gene through HR-mediated gene conversion and TLS-mediated somatic hypermutation [21]. In the initiation process of *IgV* diversification, activation-induced deaminase (AID) leads to conversion of dC to dU on regions of single-stranded DNA, the exposure of which is locally facilitated by the histone variant H3.3 [22], probably because nucleosome cores containing H3.3 are more unstable than those containing canonical H3 variants [23]. The uracil DNA glycosylase (UNG) efficiently excises the uracil residues, yielding a high frequency of abasic sites [24,25] within a defined window of around 500 base pairs. The restriction of AID-mediated abasic sites to the *IgV* gene is mediated by the *cis*-acting sequence referred to as *DIVAC* (diversification activator) [26]. The replication stalling at these abasic sites either induces template switch-mediated gene conversion with one of the 25 copies of upstream *IgV* pseudogenes carrying ~10% mismatch, resulting in HR-mediated diversification/mutagenesis of the immunoglobulin gene, or they induce TLS, resulting in somatic hypermutation at the dC/dG base pairs [14,15,24,27,28,29,30,31] (Figure 2). The induced hypermutations are most likely a consequence of replication bypass across abasic sites, rather than bypass of dU, as evidenced by the fact that most mutations are dC/dG to dG/dC transversions and loss of UNG instead leads to replication over the dU, resulting in dC/dG to dT/dA transitions [24].

The avian DT40 cell line derived from bursal B cells continues to diversify its *IgV* gene by gene conversion and somatic hypermutation during in vitro passage [32]. This cell line exhibits extraordinarily high gene-targeting efficiency, as an exception among vertebrate cell lines, and it is thus used in genetic studies due to the relative ease of establishing mutants [33]. Using this cell line, a number of studies have assessed the contribution of genes of interest to HR-mediated template switch and TLS via analyzing *IgV* diversification mechanisms [19]. Herein, we discuss how the DT40 cell line provides an opportunity to examine HR-mediated template switch and TLS at abasic sites via sequencing the constitutively diversifying *IgV* gene.

## 2. *Immunoglobulin variable* (*IgV*) Gene Diversification in DT40 Cells

To examine *IgV* sequence diversification within the limited duration of in vitro passage (usually 2–5 weeks), the following methods increase the frequency of *IgV* sequence diversification and allow effective measurement of the *IgV* diversification rate.

The first method is treating DT40 cells with a histone deacetylase inhibitor, trichostatin A (TSA). This method increases the frequency of HR-mediated gene conversion 50- to 100-fold, with limited effects on TLS-mediated somatic hypermutation [17,34]. Thus, for evaluating HR efficiency, TSA treatment during in vitro passage of DT40 cells allows for sensitive detection of gene conversion events. The mechanism underlying TSA-mediated activation of gene conversion has not been fully elucidated, but a likely scenario is that TSA mainly affects the chromatin in the upstream *IgV* pseudogenes, favoring their open configuration. Subsequently, the HR machinery can efficiently induce gene conversion using these pseudogenes as donor sequences. This view is supported by the observation that tethering of the heterochromatin protein HP1 to upstream *IgV* pseudogene regions diminishes histone acetylation within this region and alters the outcome of *IgV* diversification, causing somatic hypermutation to outweigh gene conversion [35]. The efficiency of gene conversion events can be estimated without conducting sequence analysis. Specifically, the status of surface immunoglobulin M (sIgM) expression in DT40 cl-18 cells, which carry a frameshift mutation in the *IgVλ* segment, alters from negative to positive (sIgM gain) by gene conversion-mediated replacement of the frameshift mutation. Thus, by measuring the rate of sIgM gain, the frequency of gene conversion can be estimated (Figure 3A). The percentage of sIgM-positive cells within expanding subclones can be analyzed by fluorescence activated cell sorting (FACS) [36].

Another method is AID overexpression during in vitro passage of DT40 cells. In this approach, mouse AID (mAID) is overexpressed via infection with retrovirus containing the *mAID* gene, followed by an internal ribosomal entry site and the *GFP* gene, as described previously [37]. With this method, the expression level of the mouse AID protein is approximately 20 times higher than that of endogenous chicken AID [38]. With the same methodology, chicken AID cannot be efficiently overexpressed, presumably due to protein level control (author’s unpublished data). A variant of this method is to establish clones overexpressing an enzymatically hyperactive form of human AID, hAIDup 7.3 [39], which exhibits increased levels of deamination while generating an identical pattern of deamination to the wild-type enzyme [22,39].

The overexpression of AID increases gene conversion at a rate similar to that of gene conversion induced by TSA [38]. Moreover, somatic hypermutation is significantly increased by AID overexpression, and hypermutation events are observed in nearly all *IgV* sequences analyzed, while somatic hypermutation is barely detectable without AID overexpression. AID overexpression induces gene conversion and somatic hypermutation at a ratio of approximately 1:2. AID overexpression during in vitro passage of DT40 cells allows sensitive detection of gene conversion and somatic hypermutation.

An alternative approach to analyze hypermutation events without AID overexpression has also been used. Instead of increasing the frequency of diversification, cells with spontaneously diversified *IgV* sequences can be selected by the status of sIgM expression using the cell sorter. Since the hypermutation-mediated insertion of deleterious mutations results in the loss of sIgM expression, starting from sIgM positive cells and selecting for sIgM negative cells is proposed to be a good readout of the hypermutation events. In support of this hypermutation-mediated sIgM loss is the observation that complete deletion of upstream pseudogenes (*ψV*^−^) diminishes gene conversion while increasing hypermutation events and the frequency of sIgM loss [28]. However, the caveat with this approach is that sIgM loss can also derive in good part from gene conversion-associated frameshift in wild type (WT) cells [40]. Thus, perhaps a good compromise in this regard would be to use a cell line with complete deletion of upstream pseudogenes (*ψV*^−^) and to approximate the sIgM loss with the frequency of somatic hypermutation events (Figure 3B).

Importantly, the hypermutation analysis allows evaluation not only of TLS efficiency but also of the mutation spectrum. In wild type cells, dG to dC transversion predominantly occurs [24], while in certain cases when TLS polymerase(s) are inactivated, the mutation spectrum is altered, possibly due to a shift of used TLS polymerase(s) in the bypass of replication across abasic sites in the *IgV* gene [14,27,41].

Differentiating between hypermutation and gene conversion from the *IgV* nucleotide sequence should be carried out as previously described [40]. Briefly, all sequence changes are assigned to one of the following three categories: hypermutation, gene conversion or ambiguous mutation. A single base substitution can be categorized as hypermutation or ambiguous mutation. Using the genome database search, if one can find pseudogenes identical to the sequence containing the point mutation, such mutation is categorized as ambiguous mutation, while if no identical donor sequence can be identified, such point mutation should be categorized as hypermutation. An event with multiple mutations having an identical pseudogene is interpreted as a single gene conversion event. If a sequence containing multiple substitutions is not completely identical to one of the pseudogenes, but has few mismatches, such mutations can be interpreted as deriving from a single gene conversion event followed by several subsequent hypermutations.

## 3. Templated Mutagenesis by Gene Conversion

Ig gene conversion is mediated through HR as evidenced by the fact that deletion of upstream homeologous *IgV* pseudogenes completely abolishes gene conversion [28] (Figure 4, Table 1, *Pseudo V*). This observation also demonstrates that gene conversion and hypermutation are initiated from a common DNA lesion mediated by AID, since loss of upstream *IgV* pseudogenes activates AID-dependent hypermutation (Figure 4, Table 1). This idea is further strongly supported by the fact that in DT40 cells mutated in Rad51 paralog genes *BRCA1* or *BRCA2*, in which recruitment of the Rad51 DNA-strand exchange protein to damaged DNA is strongly impaired [40,42,43], gene conversion is critically reduced and the *IgV* diversification is strongly shifted toward hypermutation (Figure 4, Table 1). Inactivation of *RAD54* (a gene encoding a key component of homologous recombination) or *FANCD2* (a gene responsible for Fanconi anemia, FA) reduces gene conversion without increasing hypermutation [44,45] (Figure 4, Table 1). These pathways might contribute to gene conversion after strand exchange, when the commitment towards Ig gene conversion has been made.

Ig gene conversion is initiated at the AID-mediated lesion (Figure 4, Table 1). HR mediates a template switch to a homeologous upstream *IgV* pseudogene. HR is also accompanied by DNA synthesis, which is facilitated by TLS-polymerases, as evidenced by the following observations. Polη-deficient (*POLη*^−/−^) cells show a reduced gene conversion rate with increased length of gene conversion tract in comparison with wild type cells [46] (Table 1). A role for Polη in D-loop extensions during HR reactions in vitro has been reported [47,48,49,50]. These data indicate involvement of Polη in Ig gene conversion and further imply that some backup polymerase(s) carrying higher processivity might compensate for the absence of Polη in *POLη*^−/−^ cells. Such backup polymerases might be Polν and Polθ, since Ig gene conversion is completely abolished in *POLη*^−/−^/*POLν*^−/−^/*POLθ*^−/−^ cells [17] (Figure 4, Table 1). Thus, these TLS polymerases play roles not only in TLS but also in HR-mediated Ig gene conversion. The recruitment of TLS polymerases is promoted by ubiquitination of the proliferating cell nuclear antigen (PCNA) [51,52,53]. However, the role of PCNA ubiquitination in vertebrate HR and *IgV* diversification has not been fully elucidated.

The heterotrimeric checkpoint clamp, consisting of the Rad9, Hus1, and Rad1 subunits (hereafter referred to as 9-1-1), is structurally similar to PCNA and serves as a DNA damage sensor to mediate the ATR signal axis [54,55,56]. The 9-1-1 clamp is loaded onto DNA by Rad17-RFC (replication factor C) in vitro, which is analogous to the PCNA-RFC clamp loader system [57,58]. In vertebrates, Rad9 and Rad17 play significant roles in HR as previously described [59,60]. *RAD9*^−/−^ and *RAD17*^/−^ DT40 cells show strongly reduced Ig gene conversion and increased hypermutation [38] (Figure 4, Table 1). Considering limited effects on HR-mediated gene-targeting and DSB repair in comparison with severe defects in Ig gene conversion by the loss of Rad9 or Rad17, the role of these factors might be confined to a subset of the HR reaction initiated by replication fork stalling and associated with formation of single strand DNA (ssDNA) gaps, rather than induced by DSBs.

Recently, mutations in the XPD family helicase gene *DDX11* were identified in the hereditary disorder Warsaw Breakage Syndrome (WABS) [61,62,63,64]. Cells from WABS patients have a shared feature with Fanconi anemia (FA) with respect to the hypersensitivity to interstrand-crosslink (ICL) inducing agents and increased chromosome aberrations, but to date *DDX11* has not been formally classified as an FA gene [61,62,63,64]. A recent study revealed that DDX11 plays a critical backup role for the FA pathway regarding ICL repair in DT40 cells [65]. In addition to its role in ICL repair, DDX11 acts jointly with the 9-1-1 clamp to promote HR induced by various lesions. *DDX11*^−/−^ DT40 cells are epistatic with *RAD17*^/−^ with regards to DNA damage hypersensitivity and induction of sister-chromatid exchanges and chromosome aberrations [65]. Interestingly, *DDX11*^−/−^ DT40 cells show reduced Ig gene conversion and hypermutation, with the effect of DDX11 in hypermutation being clearly observed in a *rad17* defective background [65] (Figure 4, Table 1). Thus, DDX11 may be involved in the promotion of both HR and TLS for releasing stalled replication forks at lesions, with its function in these two modes possibly being regulated by 9-1-1 and its loader, Rad17. This dual role of DDX11 in affecting both gene conversion and hypermutation is reminiscent of other FA genes [66] known to integrate HR and TLS machineries to promote ICL repair. However, the interplay between the FA pathway as delineated so far, and DDX11 at abasic sites, remains to be elucidated by future studies.

SPARTAN, a ubiquitin-PCNA-interacting regulator, plays a complex role in regulating mutagenesis and in mediating repair of DNA-protein crosslink lesions. On one hand, SPARTAN promotes TLS by recruiting Polη to DNA lesions, which involves its association with ubiquitinated PCNA via its PCNA interacting peptide (PIP) domain and ubiquitin-binding zinc-finger 4 (UBZ4) domain [67,68,69,70,71]. On the other hand, SPARTAN recruits the p97 protein segregase and removes Polη from DNA damage sites, thereby preventing mutations [72,73]. In addition to its function as a ubiquitinated PCNA-binding TLS regulator, SPARTAN also mediates the repair of DNA-protein crosslink lesions through its DNA-dependent metalloprotease activity [74,75,76,77]. Thus, the role of SPARTAN at the stalled replication fork has been a matter of debate and remains controversial. Recent analysis using DT40 cells revealed that SPARTAN promotes not only TLS-mediated Ig hypermutation, but also Ig gene conversion [78] (Figure 4, Table 1). For promotion of Ig gene conversion, UBZ4 and not the PIP domain of SPARTAN is required, suggesting that SPARTAN recognizes ubiquitinated protein(s) other than PCNA to promote gene conversion [78]. Important questions for future studies concern the mechanisms underlying SPARTAN’s role in the promotion of Ig gene conversion. 

Moreover, as a general remark applied to most of the studies using AID overexpression to study *IgV* gene diversification, the factors under study may affect AID activity itself, a possibility that has not been formally ruled out. Recent work from the Sale Lab (Cambridge, UK) has used an ingenuous strategy to address this issue, by co-overexpressing together with AID a bacterial uracil glycosylase inhibitor and looking at whether the mutation rates induced by dU are similar to the ones observed in control cells [22]. Likely, this strategy will be applied in the future to the study of other factors and this will increase the knowledge of the mechanisms mediating AID regulation and *IgV* diversification via bypass of abasic sites.

## 4. Nontemplated Somatic Hypermutation

Bypass replication by error-prone TLS polymerases causes hypermutation after the action of AID and UNG. Studies in budding yeast have revealed that mono-ubiquitination of PCNA at lysine 164 serves as a signal for the recruitment of error-prone TLS polymerases to sites of perturbed replication [79,80]. The role of PCNA ubiquitination was examined in *ψV*^−^ DT40 cells, in which upstream *IgV* pseudogenes are completely eliminated and gene conversion events are diminished [27] (Figure 4, Table 1). The *PCNA-K164R* DT40 cells show impaired PCNA ubiquitination and exhibit hypersensitivity to a wide variety of DNA damaging agents. Moreover, Ig hypermutation is critically reduced in *PCNA-K164R* DT40 cells (Figure 4, Table 1), indicating that post-translational modification of PCNA at lysine 164 is required for efficient TLS events in *IgV* gene diversification. Rad18, an E3-ligase required for PCNA ubiquitination in yeast, is conserved in eukaryotic cells [79,81]. *RAD18*^−/−^ DT40 cells exhibit a milder reduction in cellular tolerance to DNA damaging agents and Ig hypermutation in comparison to *PCNA-K164R* cells (Figure 4, Table 1), and this mutant strain showed residual PCNA ubiquitination [27,82]. Moreover, *RAD18*^−/−^/*PCNA-K164R* double mutant cells essentially show the same phenotype as *PCNA-K164R* mutant cells [27]. These observations suggest that Rad18-mediated PCNA ubiquitination is required for TLS events, but other ubiquitin ligases seem to function as a backup for Rad18 in vertebrate cells. The mutation spectrum in *PCNA-K164R* DT40 cells is changed, and this cell line shows biased reduction of dG:dC to dC:dG transversion [27]. Similarly, the loss of Rev1 results in nearly complete loss of the dG:dC to dC:dG transversion mutation [27] (Figure 4, Table 1). Ig hypermutation requires the deoxycytidyl transferase activity of Rev1 that is dispensable for cellular tolerance to DNA damaging agents, suggesting that the transferase activity of Rev1 incorporates dC opposite to the abasic site [41]. More importantly, other types of *IgV* hypermutations are also significantly reduced in *PCNA-K164R* cells, but not in *REV1*^−/−^ cells, suggesting that PCNA ubiquitination activates other TLS polymerases apart from Rev1.

*POLη*^−/−^/*POLζ*^−/−^ DT40 cells also show reduced dG:dC to dC:dG mutations and increased dG:dC to dA:dT mutations. Moreover, this cell line has a wild type level of Ig hypermutations [14]. These observations suggest that the absence of Polη and Polζ is largely compensated by the action of other polymerase(s) possessing activity to preferentially incorporate dA opposite the abasic site. Interestingly, *POLζ*^−/−^ DT40 cells have pronounced defects in TLS across T-T (6-4) UV photoproducts, while loss of Polη critically restores this defect. Similarly, *POLη*^−/−^/*POLζ*^−/−^ DT40 cells exhibit milder DNA damage sensitivity in comparison to *POLζ*^−/−^ cells [14]. These observations suggest that actions of Polη in the absence of Polζ critically impair the functionality of TLS. The absence of both Polη and Polζ can be substituted by other polymerase(s), such as Pοlδ. Loss of PolD3, the third subunit of the Pοlδ holoenzyme, results in a five-fold reduction of Ig hypermutation efficiency [16], suggesting a possible role of Pοlδ in TLS across abasic sites (Figure 4, Table 1). This view is supported by the observation that a complete set of human Polδ holoenzyme, but not PolD3-deficient Polδ, efficiently replicates across abasic sites in vitro [16]. Moreover, inactivation of the proofreading exonuclease activity dramatically enhances TLS across abasic sites in vitro [15], suggesting that this activity counteracts bypass replication presumably due to excision of incorporated nucleotides at damaged templates. Strikingly, the expression of proofreading exonuclease activity-deficient Polδ by introducing a heteroallelic point mutation (*POLD1^+/exo−^*) rescues the mutant phenotype of *POLD3*^−/−^ cells, and the Ig hypermutation in *POLD3*^−/−^/*POLD1^+/exo−^* returns to wild type levels [15]. These observations suggest that PolD3 counteracts Pοlδ’s proofreading exonuclease activity in order to promote TLS across abasic sites during *IgV* gene diversification, and support a pivotal role of PolD3 in Polδ-mediated TLS. The loss of PolD3 in *POLη*^−/−^/*POLζ*^−/−^ DT40 cells results in synthetic lethality, and this lethality is partially suppressed by the expression of proofreading exonuclease activity-deficient Polδ [15]. Thus, TLS relying on Polη-Polζ and Polδ might participate in parallel pathways promoting cell survival.

Either loss of Rev1 (*REV1*^−/−^) or Polη-Polζ (*POL**η*^−/−^/*REV3*^−/−^) reduces dG:dC to dC:dG transversions [14,27]. This might be attributable to the loss of Polζ activity in these mutant cells, as evidenced by the following observations. DT40 *REV1*^−/−^
*REV3*^−/−^
*REV7*^−/−^ triple mutant cells show hypersensitivity to genotoxic agents as observed in each single mutant cell, indicating an epistatic relationship between these factors [83]. These observations further suggest that Rev1, Rev3 and Rev7 function cooperatively in the TLS pathway as Polζ. 

Other TLS polymerases, including Polν and Polθ, might also be involved in Ig hypermutation, since inactivation of Polν and Polθ in *POLη*^−/−^ cells significantly reduced Ig hypermutation [17] (Figure 4, Table 1). However, to fully understand the redundancy as well as the division of labor among polymerases in the TLS network further investigations are needed.

The spatiotemporal regulation of the repair of AID-induced abasic sites, that is, whether the repair/bypass of these sites happens at the level of the fork, or behind the fork, remains an issue of investigation. So far, indirect assays have been used to assess the involvement of a subset of the factors implicated in this process in regulating fork speed and gap-filling. Specifically, DNA fiber assays following UV or MMS (methylmethane sulfonate) damage can measure damage bypass events at the fork, whereas alkaline sucrose gradient sedimentation of pulse labeled cells show the efficiency of post-replicative processes [84]. For instance, *POLη*^−/−^
*POLζ^−/−^* mutants show only defects in the post-replicative gap-filling assay, whereas *POLD3^−/−^* mutants show defects only in the fork speed assay [15,16]. Thus, although these assays do not probe for events at the *IgV* locus or in conditions with increased abasic sites, these data suggest that *IgV* diversification may happen both at the fork and in the G2 phase, and may be mediated by distinct factors.

## 5. Summary and Perspective

In this review, we summarized the current understanding of mechanisms promoting recovery of arrested replication forks at damaged DNA templates as unveiled by the analysis of *IgV* gene diversification in avian DT40 cells. This cell line exhibits extraordinarily high gene-targeting efficiency and has thus been extensively used for genetic studies [33]. Moreover, this cell line provides a great opportunity to analyze gene functions in replication fork recovery and/or gap-filling at damaged templates by analyzing *IgV* gene diversification. Thus, genetic studies using the DT40 cell line might continue to contribute to the understanding of DDT mechanisms in eukaryotic cells. The genes involved in *IgV* gene diversification (gene conversion and hypermutation) are summarized in Table 1.

Genetic studies of yeast have greatly contributed to the understanding of mechanisms regulating TLS and HR-mediated template switch to recover arrested forks at damaged DNA templates [79,85,86,87,88,89]. The mono-ubiquitination on PCNA at lysine 164 by Rad6 (E2 conjugating enzyme) and Rad18 (E3 ligase) serves as a signal for promoting TLS [87], while extension of this ubiquitin to a polyubiquitin chain by Ubc13 (E2 conjugating enzyme) and Rad5 (E3 ligase) facilitates template switch [85,89]. Studies in the mouse model using *PCNA-K164R* or *RAD18*^−/−^ mutant cells demonstrated an important role of PCNA ubiquitination in the maintenance of hematopoeotic stem cells, avoidance of premature aging and DNA damage tolerance, highlighting the critical role of this pathway for genome maintenance in mammals [90,91]. Studies using DT40 cells also revealed conserved mechanisms of the Rad18-PCNA ubiquitination axis in TLS promotion and suggested the role of additional E3 ubiquitin ligases in this ubiquitination process [27,82]. In addition to promoting TLS, Rad18 functions in concert with another E3 ligase, Rnf8, and plays a significant role in HR promotion in DT40 cells [12], as it does in budding yeast [89]. The role of Rad18 in the promotion of HR in vertebrate cells occurs via a direct interaction with the RAD51 paralog Rad51C [92]. Thus, the roles of Rad18 and the mechanisms of PCNA mono-ubiquitination might be considerably more complex in higher eukaryotic cells in comparison to yeast cells. SHPRH and HLTF are identified human homologs of Rad5 [93,94,95,96], and depletion of these E3 ligases resulted in reduction of PCNA poly-ubiquitination and rendered cells hypersensitive to DNA damaging agents [93,94]. These observations highlighted the importance of PCNA poly-ubiquitination via Rad5 homologs in human cells. However, *SHPRH*^−/−^ DT40 cells showed no detectable defects in the ubiquitination of PCNA or cellular tolerance to alkylating agents and Ig diversification (Figure 4, Table 1), but they did show higher sensitivity to the topoisomerase II inhibitor, etoposide, in comparison to wild type cells [97]. Moreover, *HLTF* is missing in the chicken genome [97]. The redundancy of E3 ligases in PCNA ubiquitination might be linked to different roles of this modification in genome maintenance and genetic plasticity. Moreover, in human cells, Ubc13 (an E2 conjugating enzyme) plays a role in DSB repair. Rnf8 (E3 ligase), in complex with Ubc13, ubiquitinates histone H2A surrounding DSB sites [98,99], and then the Rnf168 (E3 ligase)–Ubc13 complex further amplifies histone ubiquitination [100] to recruit downstream repair factors, including BRCA1, that are required for the recruitment of Rad51 to DSB sites [101,102]. Similarly, *UBC13^−^*^/−^ DT40 cells show defective DSB repair due to impaired Rad51 recruitment [103]. Furthermore, Rnf8 (E3 ligase) also contributes to both Ig hypermutation and gene conversion in DT40 cells [104] (Figure 4, Table 1). Taken together, these observations suggest that the Ubc13-Rad5 axis unveiled in yeast studies cannot be directly extrapolated into higher eukaryotic cells, due to considerable redundancy in ubiquitination enzyme networks and differential functions of the key enzymes. Moreover, the role of PCNA in HR-mediated template switch has not been elucidated. Comprehensive analysis of the ubiquitination pathway is warranted to fully dissect the molecular mechanisms of the ubiquitination signal pathway in DNA damage responses.

## Figures and Tables

**Figure 1 genes-09-00614-f001:**
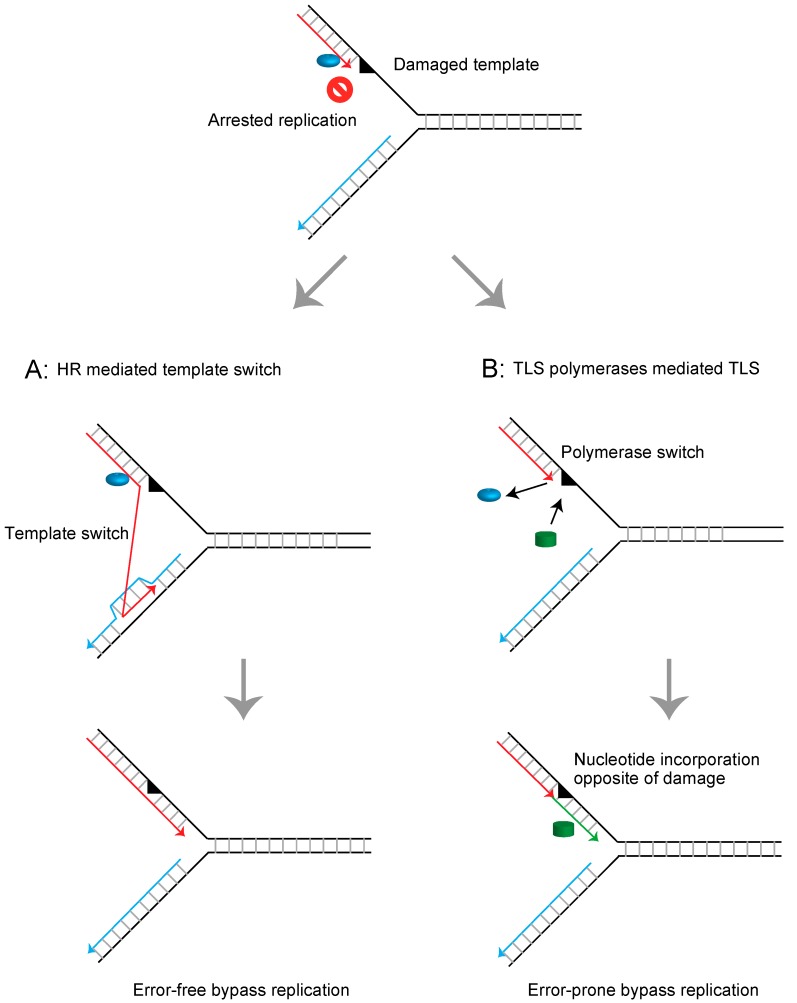
Schematic representation of mechanisms releasing the arrested replication fork at the damaged template. (**A**) Homologous recombination (HR)-mediated template switch releases the arrested replication fork using intact newly synthesized DNA as the template strand and promotes error-free bypass replication. (**B**) Translesion DNA synthesis (TLS) polymerases mediate direct bypass replication across the damaged template in an error-prone manner.

**Figure 2 genes-09-00614-f002:**
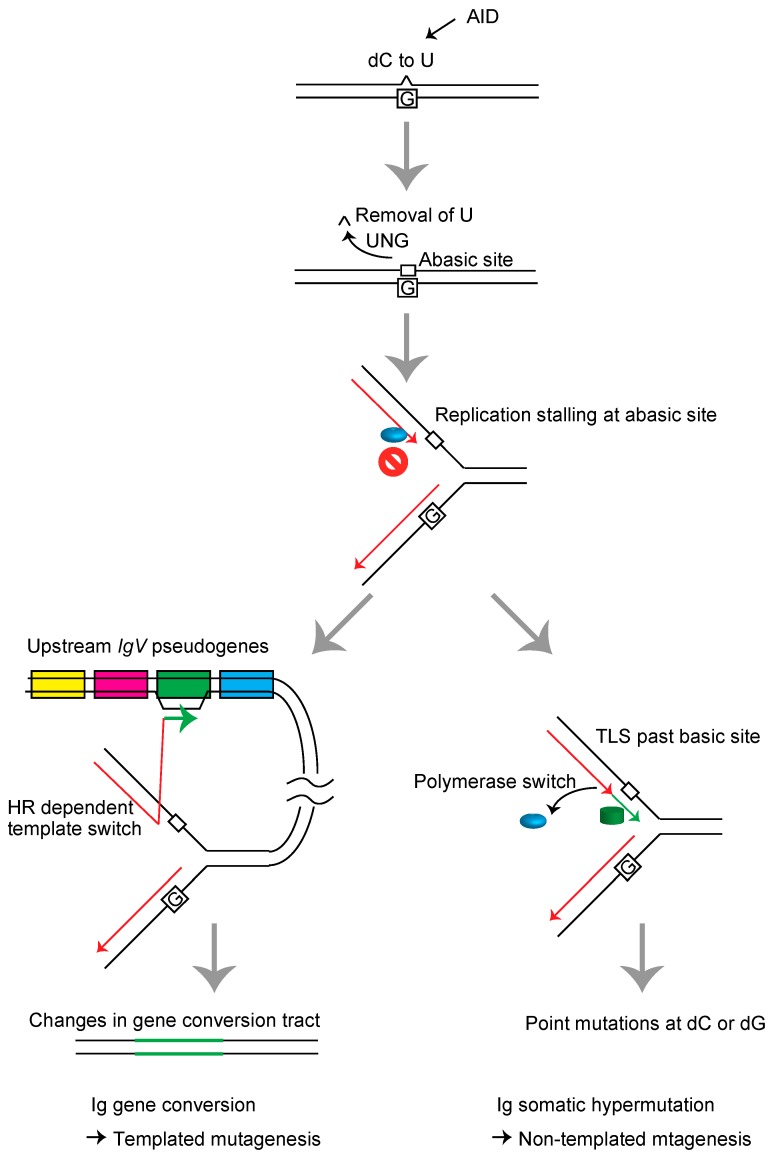
Schematic representation of the *immunoglobulin variable* (*IgV)* gene diversification mechanism in DT40 cells. The sequential actions of activation-induced deaminase (AID) and uracil DNA glycosylase (UNG) induce abasic sites in the *IgV* gene. The replication fork arrests at these lesions and induces template switch-mediated gene conversion with one of the 25 copies of upstream *IgV* pseudogenes carrying a ~10% mismatch rate, resulting in HR-mediated diversification/mutagenesis of the immunoglobulin gene (left), or TLS, resulting in somatic hypermutation at the dC/dG base pairs (right).

**Figure 3 genes-09-00614-f003:**
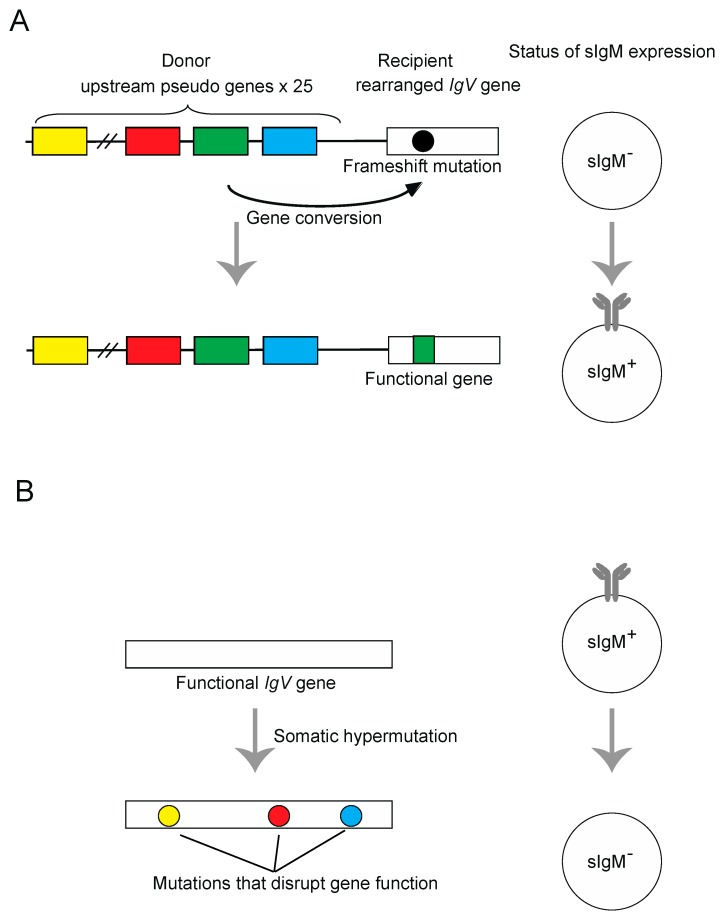
Schematic representations of the surface immunoglobulin M (sIgM) gain (**A**) and sIgM loss assays (**B**). (**A**) Principle of the Ig gene conversion assay. The sIgM-negative DT40 clone contains a frameshift in its rearranged *V-J_λ_* segments, which can be repaired by pseudogene-templated conversion events. The rate of Ig gene conversion can be measured as a gain of sIgM expression in subclones by flow-cytometric analysis of sIgM staining. (**B**) Principle of the Ig hypermutation assay. The DT40 cells carrying wild type *IgV* gene, which can be disrupted by somatic hypermutation events. The rate of hypermutation events can be measured as a loss of sIgM expression in subclones by flow-cytometric analysis of sIgM staining. The sequence analysis of hypermutation event can be also carried out by selecting sIgM negative cells by cell sorter.

**Figure 4 genes-09-00614-f004:**
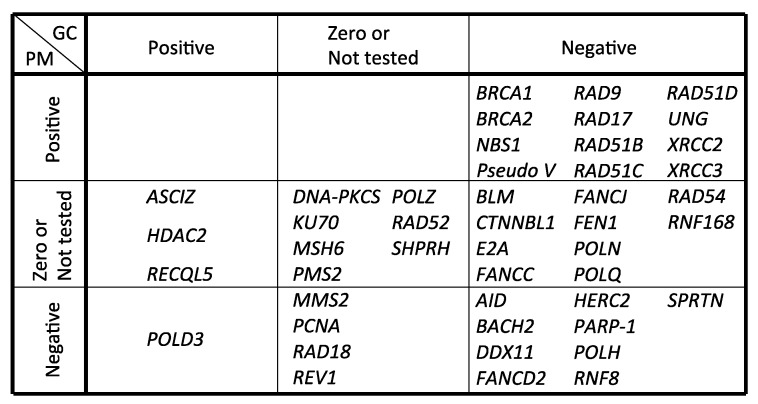
Genes involved in Ig gene conversion and hypermutation. GC and PM represent gene conversion and hypermutation, respectively. Positive, zero or negative effects by gene deletion or mutation on gene conversion and hypermutation were categorized.

**Table 1 genes-09-00614-t001:** Genes involved in Ig gene conversion and hypermutation.

Gene Name	GC	PM	Method	Reference
*AID*	−	−	IgM gain and sequencing	Arakawa et al., 2002 [105]
*ASCIZ*	+	0	IgM gain, mAID overexpression and sequencing	Oka et al., 2008 [106]
*BACH2*	−	−	IgM gain and sequencing	Budzynska et al., 2017 [107]
*BLM*	−	NT	IgM gain	Kikuchi et al., 2009 [108]
*BRCA1*	−	+	IgM loss and sequencing	Longerich et al., 2008 [42]
*BRCA2*	−	+	IgM loss and sequencing	Hatanaka et al., 2005 [43]
*CTNNBL1*	−	NT	IgM gain	Conticello et al., 2008 [109]
*DDX11*	−	−	IgM gain, mAID overexpression and sequencing	Abe et al., 2018 [65]
*DNA-PKCS*	NT	0	IgM loss	Sale et al., 2001 [40]
*E2A*	−	0	IgM gain and sequencing	Kitao et al., 2008 [110]
*FANCC*	−	NT	IgM loss and sequencing	Pace et al., 2010 [111]
*FANCD2*	−	−	IgM loss, gain and sequencing	Yamamoto et al., 2005 [45]
*FANCJ*	−	NT	IgM loss and sequencing	Kitao et al., 2011 [112]
*FEN1*	−	NT	IgM gain and sequencing	Kikuchi et al., 2005 [113]
*HDAC2*	+	0	IgM gain and sequencing	Lin et al., 2008 [114]
*HERC2*	−	−	mAID overexpression and sequencing	Mohiuddin et al., 2016 [104]
*KU70*	NT	0	IgM loss	Sale et al., 2001 [40]
*MMS2*	0	−	IgM loss and sequencing	Simpson et al., 2005 [115]
*MSH6*	0	0	IgM gain and sequencing	Campo et al., 2013 [116]
*NBS1 (p70)*	−	+	sIgM gain, mAID overexpression and sequencing	Nakahara et al., 2009 [117]
*PARP-1*	−	−	IgM gain, mAID overexpression and sequencing	Paddock et al., 2010 [118]
*PCNA (K164R)*	NT	−	IgM loss and sequencing	Arakawa et al., 2006 [27]
*PMS2*	0	0	IgM gain and sequencing	Campo et al., 2013 [116]
*POLD3*	+	−	mAID overexpression and sequencing	Hirota et al., 2015 [16]
*POLH*	−	−	IgM gain and sequencing	Kawamoto et al., 2006 [46], Kohzaki et al., 2010 [17]
*POLN*	−	0	IgM gain, mAID overexpression and sequencing	Kohzaki et al., 2010 [17]
*POLQ*	−	0	IgM gain, mAID overexpression and sequencing	Kohzaki et al., 2010 [17]
*POLH/POLN/POLQ*	−	−	IgM gain, mAID overexpression and sequencing	Kohzaki et al., 2010 [17]
*POLZ (REV3)*	0	NT	IgM gain and sequencing	Okada et al., 2005 [83]
*Pseudo V*	−	+	IgM loss and sequencing	Arakawa et al., 2004 [28]
*RAD9*	−	+	mAID overexpression and sequencing	Saberi et al., 2008 [38]
*RAD17*	−	+	mAID overexpression and sequencing	Saberi et al., 2008 [38]
*RAD18*	NT	−	IgM loss and sequencing	Arakawa et al., 2006 [27]
*RAD51B*	−	+	IgM loss and sequencing	Sale et al., 2001 [40]
*RAD51C*	−	+	IgM loss and sequencing	Hatanaka et al., 2005 [43]
*RAD51D*	−	+	IgM loss and sequencing	Hatanaka et al., 2005 [43]
*RAD52*	NT	0	IgM loss	Sale et al., 2001 [40]
*RAD54*	−	NT	IgM gain and sequencing	Bezzubova et al., 1997 [44]
*RECQL5*	+	NT	IgM gain and sequencing	Hosono et al., 2014 [119]
*RNF8*	−	−	mAID overexpression and sequencing	Mohiuddin et al., 2016 [104]
*RNF168*	−	0	mAID overexpression and sequencing	Mohiuddin et al., 2016 [104]
*SPRTN*	−	−	IgM gain, mAID overexpression and sequencing	Nakazato et al., 2018 [78]
*SHPRH*	0	0	sIgM gain and sequencing	Tomi et al., 2014 [97]
*REV1*	0	−	IgM loss and sequencing	Simpson et al., 2003 [120]
*UNG*	−	+	IgM gain, loss and sequencing	Saribasak et al., 2005 [25]
*XRCC2*	−	+	IgM loss and sequencing	Sale et al., 2001 [40]
*XRCC3*	−	+	IgM loss and sequencing	Sale et al., 2001 [40]

GC and PM represent gene conversion and hypermutation, respectively. Positive, zero and negative effects by gene deletion or mutation are shown by +, 0 and −, respectively. NT: not tested.

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
