# Peer review of "DNA Damage Tolerance Mechanisms Revealed from the Analysis of Immunoglobulin V Gene Diversification in Avian DT40 Cells"

_genes, 2018, doi:10.3390/genes9120614_

Round 1
Reviewer 1 Report
The study of DNA damage tolerance mechanisms is relevant to understanding the sources of mutagenesis. Immunoglobulin diversification in DT40 cells serves as a unique natural assay system for this purpose, for two reasons. First, it provides localised and characterised genomic lesions. Second, it provides a method for detecting homology-driven bypass events. These are normally undetectable error-free events elsewhere in the genome that use the sister chromatid as a template, but in avian Ig loci gene conversion to homeologous Ig pseudogenes provides detectable, mutagenic HR events. Immunoglobulin diversification has therefore been extensively used as a model for DNA damage tolerance, and it is timely and important to summarise the obtained results in a review.
In general, the review is well written, and will be useful for those working in the field. A simplified view is provided of the bypass pathways, but they have been amply reviewed before, partly by the authors. This is followed by a description of experimental techniques, and a clustering of results by their relevance to gene conversion or somatic hypermutation.
Especially useful is Table 1, which presents a comprehensive list and reference list of all mutants in which Ig mutagenesis has been measured. I wonder if this dataset could also be presented in a figure, for example by showing gene names on a 3x3 grid based on whether they have positive, zero or negative effects on GC or PM?
Specific comments are provided below.
Line 55, references to generating Ig diversity: Dr Sale’s 2009 review is more relevant than the ones cited, focusing on Ig diversification in DT40, albeit from the angle of the timing of bypass. It should be added: doi: 10.1098/rstb.2008.0197
Line 57-58: It might be worth discussing the issue whether all generated mutations are really the consequence of the bypass of abasic sites, and not the bypass of uracil. The cited references provide evidence for this, but the issue should be noted.
Line 79 onwards:
The article discusses methods of increasing the frequency of IgV diversification. An alternative approach has been used by many of the cited publications. Instead of increasing the frequency of diversification, cells with spontaneously diversified immunoglobulins are selected by surface IgM cell sorting, as cells with sIgM loss are good candidates for finding Ig mutation events by sequencing (as shown in Fig. 3B). This should be mentioned, possibly with a comment on the relative merits of the different approaches. The statement in line 114 “Thus, for investigating TLS functionality as well as the relationship
115 between gene conversion and hypermutation, analysis of the IgV sequence diversification is
116 conducted under AID overexpression conditions.” does not hold true for many publications cited in Table 1.
Line 153-178: The discussion of the roles of RAD9 and RAD17, and the attention drawn to DDX11, present good examples of recent results with the DT40 Ig diversification assay.
Line 163: “XPD helicase gene” should be “XPD family helicase gene”
Line 211: Calling REV1 an error-prone polymerase is misleading, as it is only really a deoxycytidyl transferase, therefore error rates cannot meaningfully measured.
Line 276-280: perhaps mention the lack of an HLTF ortholog in chickens.
Author Response
Reviewer1
The study of DNA damage tolerance mechanisms is relevant to understanding the sources of mutagenesis. Immunoglobulin diversification in DT40 cells serves as a unique natural assay system for this purpose, for two reasons. First, it provides localised and characterised genomic lesions. Second, it provides a method for detecting homology-driven bypass events. These are normally undetectable error-free events elsewhere in the genome that use the sister chromatid as a template, but in avian Ig loci gene conversion to homeologous Ig pseudogenes provides detectable, mutagenic HR events. Immunoglobulin diversification has therefore been extensively used as a model for DNA damage tolerance, and it is timely and important to summarise the obtained results in a review.
In general, the review is well written, and will be useful for those working in the field. A simplified view is provided of the bypass pathways, but they have been amply reviewed before, partly by the authors. This is followed by a description of experimental techniques, and a clustering of results by their relevance to gene conversion or somatic hypermutation.
Especially useful is Table 1, which presents a comprehensive list and reference list of all mutants in which Ig mutagenesis has been measured. I wonder if this dataset could also be presented in a figure, for example by showing gene names on a 3x3 grid based on whether they have positive, zero or negative effects on GC or PM?
(Response)
Thank you for your kind comments. According to this suggestion, we created a new figure 4 representing genes names analyzed for the involvement in Ig gene diversification.
Specific comments are provided below.
Line 55, references to generating Ig diversity: Dr Sale’s 2009 review is more relevant than the ones cited, focusing on Ig diversification in DT40, albeit from the angle of the timing of bypass. It should be added: doi: 10.1098/rstb.2008.0197
(Response)
Thank you for this critical advice. We are now citing this review article.
Line 57-58: It might be worth discussing the issue whether all generated mutations are really the consequence of the bypass of abasic sites, and not the bypass of uracil. The cited references provide evidence for this, but the issue should be noted.
(Response)
Thank you for this constructive advice. We newly noted this issue in the last part of page 5
Line 79 onwards:
The article discusses methods of increasing the frequency of IgV diversification. An alternative approach has been used by many of the cited publications. Instead of increasing the frequency of diversification, cells with spontaneously diversified immunoglobulins are selected by surface IgM cell sorting, as cells with sIgM loss are good candidates for finding Ig mutation events by sequencing (as shown in Fig. 3B). This should be mentioned, possibly with a comment on the relative merits of the different approaches. The statement in line 114 “Thus, for investigating TLS functionality as well as the relationship
115 between gene conversion and hypermutation, analysis of the IgV sequence diversification is
116 conducted under AID overexpression conditions.” does not hold true for many publications cited in Table 1.
(Response)
Thank you for this advice. We agree with this comment and added the descriptions about the method to select cells with spontaneously diversified immunoglobulin loci using the cell sorter. We also omitted the following description. ‘Thus, for investigating TLS functionality as well as the relationship between gene conversion and hypermutation, analysis of the IgV sequence diversification is conducted under AID overexpression conditions.’
Line 153-178: The discussion of the roles of RAD9 and RAD17, and the attention drawn to DDX11, present good examples of recent results with the DT40 Ig diversification assay.
Line 163: “XPD helicase gene” should be “XPD family helicase gene”
(Response)
Thank you. We corrected this error.
Line 211: Calling REV1 an error-prone polymerase is misleading, as it is only really a deoxycytidyl transferase, therefore error rates cannot meaningfully measured.
(Response)
Thank you. We corrected this error by omitting use of term ‘an error-prone polymerase’ here.
Line 276-280: perhaps mention the lack of an HLTF ortholog in chickens.
(Response)
Thank you. We now mention about this issue.
Reviewer 2 Report
In this well-written review, Abe and colleagues discussed the utility and recent findings by using the DT40 avian B lymphocyte cell line to study the DNA damage tolerance process. The DT40 model system has a long history of been used in genetic studies in metazoans due to the ease of genetic manipulation. This model system has lead to many discoveries in DNA damage tolerance field and has been used extensively to test the mechanisms identified in yeast studies.
In this short review, the authors presented methods of inducing DNA damages in DT40 cells and how to test the effects of DNA damage tolerate genes in B-cell development. The authors also covered recent discoveries made using DT40 cells on the topic of how different DNA damage factors determine the development of Ig genes.
In conclusion, I very much enjoyed reading this review and I believe that it provides an updated view of how DNA damage repair factors regulate the gene diversification at immunoglobulin locus in DT40 cells. These researches certainly shed light into the mechanisms of B-cell maturation in mouse and humans.
I recommend publication in Genes after adding a few related citations and elaborating a small section of the manuscript.
1. This paper explained why DNA damage tolerance is the best option for cell survival when lesion is encountered during replication. (PMID: 28208741)
2. line 120-124: please elaborate what exactly is this assay testing.
3. Figure 3 needs detailed explaination.
4. AID/UNG induced lesions could eventually leads to formation of DSB and triggers gene conversion (PMID: 27701075)
5. These two papers talked about PCNA-ub in mouse models (PMID: 28761001 and PMID: 26883629
6. This paper talked about the role of Rad18 in HR (PMID: 19396164)
Author Response
In this well-written review, Abe and colleagues discussed the utility and recent findings by using the DT40 avian B lymphocyte cell line to study the DNA damage tolerance process. The DT40 model system has a long history of been used in genetic studies in metazoans due to the ease of genetic manipulation. This model system has lead to many discoveries in DNA damage tolerance field and has been used extensively to test the mechanisms identified in yeast studies.
In this short review, the authors presented methods of inducing DNA damages in DT40 cells and how to test the effects of DNA damage tolerate genes in B-cell development. The authors also covered recent discoveries made using DT40 cells on the topic of how different DNA damage factors determine the development of Ig genes.
In conclusion, I very much enjoyed reading this review and I believe that it provides an updated view of how DNA damage repair factors regulate the gene diversification at immunoglobulin locus in DT40 cells. These researches certainly shed light into the mechanisms of B-cell maturation in mouse and humans.
I recommend publication in Genes after adding a few related citations and elaborating a small section of the manuscript.
1. This paper explained why DNA damage tolerance is the best option for cell survival when lesion is encountered during replication. (PMID: 28208741)
(Response)
Thank you for this kind advice. We added the suggested reference.
2. line 120-124: please elaborate what exactly is this assay testing.
(Response)
Thank you for this kind advice. We revised these sentences to exactly describe what is this assay testing.
3. Figure 3 needs detailed explanation.
(Response)
We added descriptions to explain these assays.
4. AID/UNG induced lesions could eventually leads to formation of DSB and triggers gene conversion (PMID: 27701075)
(Response)
Thank you for this comment. We are a little confused with this comment. The suggested study artificially induced DSBs using I-SceI endonuclease and found Ig gene conversions as well as In/del events replaced cl-18 mutation. However, this article showed no direct evidence indicating that AID induces DSB in physiological condition. We believe that replication arrests at the abasic sites (mediated by AID) are the shared substrate for Ig gene conversion and hypermutation in physiological condition as evidenced by the observations that HR mediated Ig gene conversion is reduced in HR mutant cells (xrcc3, rad9, and rad17 cells) which instead exhibit increased number of hypermutation events (Saberi et al 2008 MCB).
5. These two papers talked about PCNA-ub in mouse models (PMID: 28761001 and PMID: 26883629
(Response)
Thank you. We cited these references in the discussion part.
6. This paper talked about the role of Rad18 in HR (PMID: 19396164)
(Response)
Thank you. We cited the suggested reference in the discussion part.
Reviewer 3 Report
The manuscript entitled “DNA damage tolerance mechanisms revealed from the analysis of Immunoglobulin V gene diversification in avian DT40 cells” by Abe T et al, is a very well written manuscript.
The manuscript covers the certain aspects of DNA damage tolerance in avian DT40 cells, largely focusing on gene conversions and somatic hypermutations. In this elegantly written manuscript authors have concisely explained the DNA damage tolerance mechanisms in avian DT40 cells.
The review manuscript provides substantial amount information on the translesion DNA synthesis and its role DNA damage tolerance. The roles of proof-reading and also error-prone DNA polymerases in DNA damage tolerance are adequately explained.
I have a very few minor comments to make.
Comment # 1
In line # 133, the Rad51 is described as a recombinase. Although, in the literature, the Rad51 is as depicted as recombinase. In my opinion, it is not a recombinase! It is actually (in true sense!) a DNA-strand exchange protein. So, I would like authors to replace the word “recombinase” with “DNA-strand exchange protein”.
Recombinase is the one that performs recombination process single-handedly, for ex. Cre recombinase. However, the Rad51 performs the homologous dependent DNA strand exchange function. So, it should be termed as a DNA-strand exchange protein.
Comment # 2
In line 135. Rad54 is mentioned as “a gene involved in later steps of homologous recombination”. Actually, as a chromatin demodulator, it also plays an important role in the initial steps of homologous recombination by removing (or by sliding) histones, a prerequisite step for the DNA resection. Therefore, the sentence “a gene involved in later steps of homologous recombination” can be replaced with “a key component of homologous recombination”.
Comment # 3
In line #150 -151 the word “cell” is missing between the “proliferating and nuclear” words. Please write it as “Proliferating cell nuclease antigen (PCNA)”.
Author Response
Comment # 1
In line # 133, the Rad51 is described as a recombinase. Although, in the literature, the Rad51 is as depicted as recombinase. In my opinion, it is not a recombinase! It is actually (in true sense!) a DNA-strand exchange protein. So, I would like authors to replace the word “recombinase” with “DNA-strand exchange protein”.
Recombinase is the one that performs recombination process single-handedly, for ex. Cre recombinase. However, the Rad51 performs the homologous dependent DNA strand exchange function. So, it should be termed as a DNA-strand exchange protein.
(Response)
Thank you. We omitted the use of term ‘recombinase’ and replaced the word with “DNA-strand exchange protein”.
Comment # 2
In line 135. Rad54 is mentioned as “a gene involved in later steps of homologous recombination”. Actually, as a chromatin demodulator, it also plays an important role in the initial steps of homologous recombination by removing (or by sliding) histones, a prerequisite step for the DNA resection. Therefore, the sentence “a gene involved in later steps of homologous recombination” can be replaced with “a key component of homologous recombination”.
(Response)
Thank you. We changed this sentence according to this kind advice.
Comment # 3
In line #150 -151 the word “cell” is missing between the “proliferating and nuclear” words. Please write it as “Proliferating cell nuclease antigen (PCNA)”.
(Response)
Thank you. We corrected the typological error. Thank you very much for this kind advice.
Reviewer 4 Report
In this review paper, the authors summarized the studies of gene conversion and somatic hypermutation that occur at the IgV locus in avian DT40 cells. The paper is well written and instructive. But, I have several concerns listed as follows:
Major points:
DNA damage is induced on the template strand of the lagging strand synthesis in Figure 1. I was confused to see that abasic site is created on the template strand of the leading strand synthesis in Figure 2. Could the authors explain the reason for illustrating them in opposite ways?
The authors mentioned the effect of histone H3.3 (lines 56-57). Could the authors briefly explain how H3.3 affects the exposure of single-stranded DNA.
As AID induces IgV diversification, the readers would appreciate if the authors could explain the regulation of AID. For example, how the action of AID is limited to B cells and why dC to dU is restricted to a defined window of 500 base pair (lines 58-59).
As the IgV gene constitutively diversifies (lines 69-79), each IgV gene suffers multiple rounds of gene conversion and/or somatic hypermutation? Alternatively, the IgV gene suffers only one round of gene conversion or somatic hypermutation? Could the authors explain how one can interpret the sequencing data of the IgV gene and discriminate (or count) gene conversion and somatic hypermutation events, in the section ‘2. IgV diversification in DT40 cells’.
The authors argue that ‘This, by starting from sIgM positive cells … somatic hypermutation frequency can be estimated (Figure 3B). (lines 123-124). However, this is the case only when none of the IgV pseudogenes contain mutations that cause sIgM loss. Could the authors explain more details.
The authors explained that DDX11 and SPARTAN are required for both gene conversion and hypermutation (page 6). Isn’t it possible that DDX11 and SPARTAN affect the initial DNA events such as deamination, which induce either gene conversion or hypermutation. Could the authors give some comments on this.
Either loss of Rev1 or PolηPolζ reduces dG:dC to dC:dG transversions (page 7). Could the authors explain the relationship between Rev1 and PolηPolζpolymerases.
Minor points:
It is better to change ‘AID (line 26)’ to ‘activation-induced deaminase (AID)’.
To make it clear, it may be better to change ‘[24](Table1)’ to ‘[24](Table1, Pseudo V)’ (line 129)’.
‘the deoxycytidyl activity transferase activity of Rev1 (line 212)’ must be ‘the deoxycytidyl transferase activity of Rev1’
‘a pivotal of role of PolD3 (line 236)’ can be ‘a pivotal role of PolD3’.
In the reference list, not all but many papers include DOI. Please follow the instruction.
It is better to explain what ‘GC’ and ‘PM’ stand for in the table legend.
Author Response
In this review paper, the authors summarized the studies of gene conversion and somatic hypermutation that occur at the IgV locus in avian DT40 cells. The paper is well written and instructive. But, I have several concerns listed as follows:
Major points:
DNA damage is induced on the template strand of the lagging strand synthesis in Figure 1. I was confused to see that abasic site is created on the template strand of the leading strand synthesis in Figure 2. Could the authors explain the reason for illustrating them in opposite ways?
(Response)
We revised the figure 1, so as to avoid this confusion. In the new figure 1, DNA damage is located on the template for leading strand synthesis.
The authors mentioned the effect of histone H3.3 (lines 56-57). Could the authors briefly explain how H3.3 affects the exposure of single-stranded DNA.
(Response)
Thank you. We expanded on the conclusions drawn in the Romanello et al, 2016 study on the roles of H3.3 in supporting AID function.
As AID induces IgV diversification, the readers would appreciate if the authors could explain the regulation of AID. For example, how the action of AID is limited to B cells and why dC to dU is restricted to a defined window of 500 base pair (lines 58-59).
(Response)
We added the descriptions about the mechanism how AID’s action is restricted to IgV gene.
As the IgV gene constitutively diversifies (lines 69-79), each IgV gene suffers multiple rounds of gene conversion and/or somatic hypermutation? Alternatively, the IgV gene suffers only one round of gene conversion or somatic hypermutation? Could the authors explain how one can interpret the sequencing data of the IgV gene and discriminate (or count) gene conversion and somatic hypermutation events, in the section ‘2. IgV diversification in DT40 cells’.
(Response)
To comply with this comment, we described the method how the gene conversion and hypermutation are interpreted from the sequence data.
The authors argue that ‘Thus, by starting from sIgM positive cells … somatic hypermutation frequency can be estimated (Figure 3B). (lines 123-124). However, this is the case only when none of the IgV pseudogenes contain mutations that cause sIgM loss. Could the authors explain more details.
(Response)
We now clarified this issue in the text (the first paragraph in page 10).
The authors explained that DDX11 and SPARTAN are required for both gene conversion and hypermutation (page 6). Isn’t it possible that DDX11 and SPARTAN affect the initial DNA events such as deamination, which induce either gene conversion or hypermutation. Could the authors give some comments on this.
(Response)
We have unpublished results that AID-induced lesions occur with similar frequency in ddx11 cells, but this issue requires more investigations before formal publication. We have commented in the manuscript about the possibility that AID activity is reduced by some of the mutations reducing IgV diversification, which is left unexplored by most of the studies (the first paragraph in page 17).
Either loss of Rev1 or PolηPolζ reduces dG:dC to dC:dG transversions (page 7). Could the authors explain the relationship between Rev1 and PolηPolζpolymerases.
(Response)
Thank you for this comment. We added the explanations about the genetic relationship between these polymerases and Rev1 as follows.
‘Either loss of Rev1 (REV1-/-) or Polη-Polζ (POLη-/-/REV3-/-) reduces dG:dC to dC:dG transversions. This might be attributable to the loss of Polζ activity in these mutant cells as evidenced by the following observations. DT40 REV1−/− REV3−/− REV7−/− triple mutant cells showed hypersensitivity to genotoxic agents as observed in each single mutant cells, indicating an epistatic relationship between these factors. These observations further suggest that Rev1, Rev3 and Rev7 play cooperative roles in the TLS pathway, as Polz. ’
Minor points:
It is better to change ‘AID (line 26)’ to ‘activation-induced deaminase (AID)’.
(Response)
Thank you. We corrected this.
To make it clear, it may be better to change ‘[24](Table1)’ to ‘[24](Table1, Pseudo V)’ (line 129)’.
(Response)
Thank you. We corrected this.
‘the deoxycytidyl activity transferase activity of Rev1 (line 212)’ must be ‘the deoxycytidyl transferase activity of Rev1’
(Response)
Thank you. We corrected this.
‘a pivotal of role of PolD3 (line 236)’ can be ‘a pivotal role of PolD3’.
(Response)
Thank you. We corrected this.
In the reference list, not all but many papers include DOI. Please follow the instruction.
(Response)
Thank you. We corrected these errors.
It is better to explain what ‘GC’ and ‘PM’ stand for in the table legend.
(Response)
Thank you. We added explanations.